# Identification of Glyceraldehyde-3-Phosphate Dehydrogenase Gene as an Alternative Safe Harbor Locus in Pig Genome

**DOI:** 10.3390/genes10090660

**Published:** 2019-08-29

**Authors:** Xiaosong Han, Youcai Xiong, Changzhi Zhao, Shengsong Xie, Changchun Li, Xinyun Li, Xiangdong Liu, Kui Li, Shuhong Zhao, Jinxue Ruan

**Affiliations:** 1Key Laboratory of Agricultural Animal Genetics, Breeding and Reproduction (Huazhong Agricultural University), Ministry of Education, Wuhan 430070, China; 2The Cooperative Innovation Center for Sustainable Pig Production—Swine Breeding and Reproduction Innovation Platform, Huazhong Agricultural University, Wuhan 430070, China; 3State Key Laboratory of Animal Nutrition, Institute of Animal Sciences, Chinese Academy of Agricultural Sciences, Beijing 100193, China; 4College of Life Science, Foshan University, Guangdong 528231, China

**Keywords:** GAPDH, safe harbor, CRISPR/Cas9, homologous recombination, pig

## Abstract

The ectopic overexpression of foreign genes in animal genomes is an important strategy for gain-of-function study and establishment of transgenic animal models. Previous studies showed that two loci (*Rosa26* and *pH11*) were identified as safe harbor locus in pig genomes, which means foreign genes can be integrated into this locus for stable expression. Moreover, integration of a transgene may interfere with the endogenous gene expression of the target locus after the foreign fragments are inserted. Here, we provide a new strategy for efficient transgene knock-in in the endogenous *GAPDH* gene via CRISPR/Cas9 mediated homologous recombination. This strategy has no influence on the expression of the endogenous *GAPDH* gene. Thus, the *GAPDH* locus is a new alternative safe harbor locus in the pig genome for foreign gene knock-ins. This strategy is promising for agricultural breeding and biomedical model applications.

## 1. Introduction

Integration of exogenous genes into the genome for stable expression is an important strategy for gain-of-function study and establishment of transgenic animal models. Traditionally, ectopic overexpression of foreign genes in a genome is randomly performed [1,2,3], which means the foreign genes can be inserted anywhere in the genome. However, random integration of foreign genes often leads to unstable expression or unpredictable phenotypes [4]. Homologous recombination (HR) is an important pathway involved in the repair of double-strand DNA breaks [5], and HR-mediated gene knock-in allows the precise insertion of foreign fragments. HR-mediated integration in site-specific locus was used in the past to produce animal models [6,7]. This method is inefficient because of low efficiency [8], and laborious as construction of donor vectors is required.

Engineering site-specific nucleases, such as zinc finger nucleases (ZFN), transcription activator-like effector nucleases (TALENs), and newly developed clustered regularly interspersed short palindromic repeat (CRISPR)/CRISPR-associated protein 9 (Cas9) systems are proven to introduce double-strand breaks (DSBs) in a specific locus [9,10,11], and when supplied with a donor vector, HR efficiency can be improved greatly. CRISPR/Cas9 systems contain a single guide RNA (sgRNA) and a Cas9 endonuclease. The Cas9 protein can be introduced to the target locus under the guidance of the sgRNA. Their high efficiency and simple constructs are the reasons why CRISPR/Cas9 systems have been widely used in human, mice, zebrafish, and other species [12,13,14]. Pig models have great potential for agricultural and biomedical applications because of their high similarity to humans in physiology, and anatomy [15].

In 2014, Li and colleagues identified *Rosa26* locus as a safe harbor in the pig genome [16], and subsequently, Runa et al. identified pH11 gene as another safe harbor locus in the pig genome, and they inserted a gene fragment larger than 9 kb at the *pH11* locus via CRISPR/Cas9 technology [17]. In these studies, foreign genes were integrated into the *Rosa26* and *pH11* locus, which may interfere with these inner genes and require multiple regulatory elements in the donor vectors.

House-keeping genes are stably expressed in cells and almost all living beings. Many house-keeping genes are used as reference genes in many molecular techniques due to their stable and high expression [18]. We hypothesized that those house-keeping genes may be proper loci for integrating foreign genes.

In this study, we investigated if house-keeping gene glyceraldehyde-3-phosphate dehydrogenase (*GAPDH*) can be used as an alternative safe harbor locus. Based on the CRISPR/Cas9 expression plasmid PX330 [11], we generated vectors that target the downstream of GAPDH gene in the pig genome, and another donor vector contains 2A-GFP fragment without a promoter. We examined if 2A-GFP fragment can be expressed in frame with *GAPDH* coding sequence while the endogenous gene *GAPDH* is expressed as normal. Whereas the GFP can express under the control of *GAPDH* promoter. Our results provide an alternative strategy to integrate exogenous genes in the pig genome.

## 2. Materials and Methods 

### 2.1. Plasmids

One sgRNA targeting the GAPDH locus was designed using online software: CRISPR-offinder [19]. Oligonucleotides coding for the sgRNAs were annealed and assembled into a linearized px330 vector (addgene, #42230) according to the method described by Zhang at the Broad Institute of MIT [11]. The oligonucleotides coding for the sgRNAs were denatured using a thermocycler with the following program: 95 °C, 5 min; 65 °C, 30 min; and hold at 4 °C. Then, annealed oligos were ligated with *BbsI*-digested PX330 vector, and subsequently, the ligation mixture was transformed into *Escherichia coli* DH5α competent cells (TakaRa, Otsu, Japan). All the primer pairs and guide RNA sequences are listed in the Appendix A.

The donor vector (pCDNA3.1-GAPDH-GFP-KI-donor) was constructed using pCDNA3.1(+) as the backbone. The GFP sequences flanking with homology arms were synthesized and inserted into pCDN3.1(+) by restriction enzymes (Genscript, Nanjing, China). Both the 5′ and 3′ homology arms for HR events at GAPDH locus were 900 bp in size. Detailed donor vector sequences are listed in the Appendix A.

### 2.2. Cell Culture and Transfection

PK15 and 3D4/21 cell lines were maintained in DMEM supplemented with 10% fetal bovine serum (Hyclone, Logan, UT, USA) and 1% penicillin/streptomycin (Life Technology, Rockville, MD, USA). Primary porcine fetal fibroblasts (PFFs) were isolated from 35-day-old fetuses as previously reported [20]. The PFF cells were maintained in DME/F12 supplemented with 15% fetal bovine serum (Hyclone, Logan, UT, USA) and 1% penicillin/streptomycin (Life Technology, Carlsbad, CA, USA). All cell lines were maintained at 37 °C and 5% CO_2_ in a humidified incubator.

A day before transfection, PK15 and 3D4/21 cell lines were seeded onto 6-well plates. When the cells were 70–80% confluent, they were co-transfected with PX330 and donor vector (at ratio = 1:1) using Lipofectamine 2000 reagent (Invitrogen, Carlsbad, CA, USA) following the manufacturer’s protocol. For PFF Cells, approximately 1 × 10^6^ cells were resuspended in 200 μL electroporation buffer (Bio-Rad, California, CA, USA) and supplemented with PX330 and donor vector. Fibroblasts were transfected using the Gene Pulser Xcell^TM^ (Bio-Rad, California, CA, USA) at 175 V/20 ms. Subsequently, cells were transferred to 6-well plates. G418 (700 μg/mL Sigma St. Louis, MO, USA) was added 48 h after transfection for 3 days.

### 2.3. T7EN1 Detection Assay and Sequencing

To detect the activity of sgRNA, which targets the GAPDH locus, we performed the T7 endonuclease 1(T7EN1) assay. After transfection, cells were incubated for 48 h at 37 °C. Then, genomic DNA was extracted using TIANapm Genomic DNA kit (TIANGEN, Beijing, China). PCR amplification of the targeted region from a pool of sgRNA-treated cells generated a mixture of WT and mutant amplicons, and the PCR products were purified using TaKaRa MiniBEST DNA Fragment Purification Kit (TaKaRa, Otsu, Japan), following the manufacture’s protocol. Melting and reannealing of the purified PCR products resulted in the formation of mismatches between heteroduplexes of the WT and mutant alleles using a temperature program of 95 °C for 10 min, 95 °C to 85 °C ramping at −2 °C/s, 85 °C to 25 °C at −0.25 °C/s, and 15 °C hold for 2 min. After reannealing, products were digested with 0.5 μL T7EN1 (NEB, Ipswich, MA, UK) enzyme at 37 °C for 10 min and then analyzed on 2% agarose gel. The gels were stained with Gel-rad and quantified by densitometry using the ImageLab software (Bio-Rad, California, CA, USA). Sanger sequencing was used to further confirm the gene editing rate. The online website tool TIDE was used to calculate the indel rate [21].

### 2.4. Fluorescence-Activated Cell Sorting (FACS)

Surviving cells after G418 treated were trypsinized and collected. Then green fluorescent protein (GFP)-positive cells were sorted using the BD FACS Aria^TM^ sorting machine (BD Biosciences, Franklin Lakes, NJ, USA), analyzed with FlowJo software. GFP-positive cells were collected and expanded for Western blot and immunofluorescence assay (IFA).

### 2.5. Immunofluorescence Assay (IFA)

GFP-positive PK15 cells were seeded onto 12-well plate. When the cells are 80% confluent, they were fixing with 4% paraformaldehyde for 15 min at room temperature (RT) and then permeabilized with PBS containing 0.3% Triton X-100 for 15 min at RT. Next, the cells were blocked using 1% bovine serum albumin (BSA) in PBS for 40 min and then incubated with anti-GFP and anti-GAPDH antibody for 2 h separately. After washing thrice in PBS, the cells were incubated for 1.5 h with a goat anti-rabbit secondary antibody conjugated with Alexa Fluor 488 (Proteintech, Rosemont, IL, USA). The cells were finally counterstained with 4′,6′-diamidino-2 phenylindole (DAPI) and visualized by a fluorescent microscope (Olympus, Tokyo, Japan) at 100X magnification.

### 2.6. Western Blot Analysis

To verify the expression of GFP and GAPDH, GFP-positive PK15 cells were collected and lysed in RIPA buffer (Sigma-Aldrich, St. Louis, CA, USA), and the total proteins were quantified with a BCA kit (Beyotime, Shanghai, China). An equal amount of the total proteins from each sample was separated on 12% sodium dodecyl-sulphate polyacrylamide gel electrophoresis (SDS-PAGE) and transferred to a polyvinylidene difluoride membrane (Millipore, Darmstadt, Germany). Then, membranes were blocked with 5% non-fat milk and incubated with *GAPDH* and GFP antibody at dilution of 1:1000 (Proteintech, Rosemont, IL, USA). After washing with TBST, HRP-conjugated anti-Rabbit IgG was used as the secondary antibody. Digital signal of chemiluminescent Western blotting was acquired by Bio-Rad GelDoc XR and ChemiDoc XRS system, and analysis was conducted by the Quantity One program, version 4.6 (Bio-Rad, California, CA, USA).

### 2.7. Off-Target Analysis of sgRNA

Potential off-target sites (OTS) were predicted using an online software: CRISPR-offinder [19]. We identified the top 7 sites with the highest similarity to GAPDH-sgR. Fourteen pairs of primers were designed to amplify the potential off-target sites from the genomic DNA isolated from the GFP-sorting cells. Sanger sequencing was performed to determine whether any mutations occurred.

### 2.8. Cell Proliferation Assay

The growth properties of cells were measured by xCELLigence RTCA DP station (Roche Diagnostics, Basel, Switzerland) configured with a 16-well E-plate, according to the manufacturer’s protocol. Briefly, background measurements were taken from the wells by adding a 50 μL medium to the E-16 plates. Then, cells were seeded at a concentration of 3500 cells/well in the E-plate, and then the E-plate was incubated at room temperature for 30 min for the cells to settle and was returned to the workstation in the incubator. The cell index (CI) values were continuously displayed on the machine user interface and recorded every 15 min for 40 h.

### 2.9. Quantitative RT-PCR (qPCR) and Statistical Analysis

To investigate the levels of gene expressions after poly(I:C) stimulation, total RNA obtained from the samples was converted into cDNA using the PrimeScript^TM^ RT reagent Kit with gDNA Eraser (Perfect Real Time) (TaKaRa, Otsu, Japan). The qPCR was performed using the SYBR® Green Real-time PCR Master Mix (Toyobo Co., Ltd., Osaka, Japan) on the CFX384 Touch™ Real-Time PCR Detection System following the manufacturer’s instructions (Bio-Rad, California, USA). The Oligo7 Primer Analysis software (Molecular Biology Insights, Inc., Cascade, CO, USA) was applied to design and evaluate the primers for gene validations. The specific primer sequences used for qPCR are shown in Appendix A. The qPCR was performed as follows: 1 cycle at 95 °C for 2 min, 40 cycles at 95 °C for 30 s, 60 °C for 20 s and 72 °C for 15 s. *β-actin* (*ACTB*) was used as a control gene to normalize the expression levels. The student’s t-test was used to analyze the differential expression of genes.

## 3. Results

### 3.1. Generation of a Reporter System in Pig Genome

This study aims to identify whether the endogenous *GAPDH* gene can act as an alternative safe harbor locus in the pig genome. Thus, we constructed a reporter system targeting the *GAPDH* locus in the pig genome. One sgRNA was designed to target the 3′ end of *GAPDH* just upstream of the stop codon. To achieve GFP gene knock-in at the *GAPDH* locus, a homologous recombination vector was constructed as a donor vector (Figure 1A and Appendix A). The donor vector was generated to carry a promoterless 2A-GFP sequence flanked by two regions of homology by modifying the pCDNA3.1(+) vector. When HR-mediated knock-in events occurred, the 2A-GFP fragment was inserted in frame with the endogenous *GAPDH* coding sequence, and because the self-cleaving 2A peptide exists, the *GAPDH* and GFP can be expressed separately (Figure 1A).

For genome editing, we used CRISPR/Cas9 expression vector PX330, and one sgRNA (*GAPDH*-sgR) was introduced into the chimeric guide scaffold via *BbsI*-restriction sites to generate the PX330-GAPDH-sgR vector. The sequence was verified through Sanger sequencing. To test the activity of this sgRNA, the plasmid PX330-GAPDH-sgR was transfected into 3D4/21 cells, and the genomic DNA was extracted 48 h post transfection. Specific primers spanning the target site were designed to amply the mutated locus, and the PCR products were used to detect the activity of sgRNA by T7EN1cleavage assay. Compared with the negative control cells, the PX330-GAPDH-sgR plasmid-treated group showed apparent activity (Figure 1B). Moreover, the PCR product was directly subjected to Sanger sequencing, and corresponding to the T7EN1 results, the presence of multiple peaks after the target site in the sequencing curves clearly distinguished the mutants from the non-targeted cells (Figure 1C). Then, we estimated the frequency of mutation by the online software TIDE. The mutation percentage rate was 14.3% (Appendix A).

Potential off-target sites (OTS) were predicted by online website tool CRISPR-offinder. The potential off-target sites were aligned to the whole pig genome, and the OTSs with less than 4 mismatches to the GAPDH-sgR were selected for further analysis (Table 1). PCR-Sanger sequencing was performed, and the results showed that no mutation occurred in the potential off-target loci.

### 3.2. Quantification of HR-Mediated Knock-in in Various Pig Cells

To test whether the reporter system works in the pig genome, we co-transfected the PK15 cells with PX330-GAPDH-sgR plasmid and the donor vector (pCDNA3.1-GAPDH-GFP-KI-donor). As expected, we observed GFP+ cells 24 h after transfection, whereas no GFP+ cells were detected in the absence of either sgRNAs or Cas9. We also used the reporter system in other pig cells, including 3D4/21 and fetal fibroblast cells (PFF), and we observed green fluorescence in both of these cell lines (Figure 2A).

Due to the GFP expression, we can direct assessment of the knock-in efficiency by FACS analysis. In the three different cell lines, the average frequencies of GFP+ cells are over 2%, as follows: 2.09% in PK15 cells, 2.47% in PFF, and 2.18% in 3D4/21 cell lines (Figure 2B). To further identify the site-specific insertion of GFP, we designed two pairs of primers that span the homology arms to amplify the junctions at the two ends between the transgene fragment and genomic DNA (Figure 1A and Figure 3A). Genome PCR and sequencing analysis of the sorted GFP+ cells showed that the 2A-GFP fragment indeed integrated precisely at the 3′ end of the *GAPDH* gene (Figure 3B), which supported the fact that the targeting process was mediated by the HR pathway.

### 3.3. Identification of the Protein Expression of GAPDH Gene

The strategy of this study is to insert the GFP sequence into the downstream of the *GAPDH* locus, and the GFP can be expressed in a frame with the *GAPDH* CDS. Thus, we inferred that the endogenous *GAPDH* gene could not be disrupted. To further explore if there are some influences on the expression of *GAPDH* gene after editing, the GFP+ sorted PK15 cells were selected for analysis by immunofluorescence assay (Figure 4A) and Western blot (Figure 4B). As expected, we observed the expression of *GAPDH* in GFP+ cells and in normal PK15 cells. Comparing with wild type PK15 cells, the protein expression levels of GFP+ PK15 cells have no difference.

To detect if there was any malicious impact on the pig cells after knock-in events occurred, we performed a real time cell analyzer (RTCA) assay to compare the growth properties between the wild type PK15 cells with the GFP knock-in PK15 cells (Appendix A). And the results showed that the GFP+ PK15 cells had similar cell viability with the wild type PK15 cells.

In addition, to verify if there were any effects at the transcriptional level in the GFP knock-in PK15 cells, five genes were picked to performed RT-qPCR assay (Appendix A). Among them, two genes (*NCAPD2* and *IFF01*) are located in the either side of *GAPDH* gene, and two genes (*TP53* and *GLB1*) are regulated by *GAPDH* gene [22,23]. By applying RT-qPCR, it showed that there was no difference in the knock-in PK15 cells and wild type PK15 cells. Taken together, those results demonstrated that this method is practicable.

## 4. Discussion

In recent years, CRISPR/Cas9 systems have been applied successfully to create genetically modified animal species [24,25,26,27]. As one of the most important agricultural animals, pig is used as an ideal model for animal breeding and biomedical studies. Wang et al. used the CRISPR/Cas9 system to knockout myostatin (MSTN) in pigs, thereby increasing the amount of muscle [28]. Whitworth et al. edited the CD163 in the pig genome via CRISPR/Cas9, and the pig survived the PRRSV challenge [29]. Using CRISPR/Cas9 and somatic nuclear transfer technology, Yan et al. established a knock-in pig model of Huntington’s disease [30]. However, there are few studies on transgene pig models. As of now, two safe harbor loci have been identified in the pig genome [16,17]. Rosa26 locus encodes non-coding RNAs that are ubiquitously expressed with a strong promoter. Thus, when integrating foreign genes in this locus, it may interfere with the transgene. When integrating genes of interests in the *pH11* locus, the promoter needs to be added upstream of the transgene fragments. Apart from that, the donor vectors used in those safe harbor loci need to consist of several regulatory elements, such as splice acceptors (SA), polyA, and/or promoters. To some extent, this may be laborious.

House-keeping genes are stably expressed in cells and organizations in all living beings, and they are essential for life. Glyceraldehyde-3-phosphate dehydrogenase (*GAPDH*) gene is one of the key house-keeping genes, and this gene is ubiquitously expressed in almost all organs and tissues [31]. In many laboratory techniques, such as Western blot, QPCR, and so on, *GAPDH* is taken as a reference gene. In consideration of the characteristics of *GAPDH*, we inferred that *GAPDH* gene can be applied to be a new safe harbor locus.

In this study, we designed a sgRNA targeting the 3′ end of the *GAPDH* gene close to the stop codon and constructed a donor vector consisting of a GFP fragment. Successfully, we inserted GFP fragment in the 3′ end of the *GAPDH* gene through homologous recombination. As expected, the GFP expressed without disrupting the internal gene. These results demonstrated that the strategy we performed in this work is promising. Additionally, the GFP is driven in the control of the inner gene promoter and the transcription termination factor, thereby simplifying the construction of donor vectors. According to the strategy mentioned here, we infer that other house-keeping genes, such as *ACTB*, *B2M*, *SDHA* [18], can be applied as new types of safe harbor loci. This would broaden the use of these loci and offer more sites for integrating foreign genes.

Although the HR-mediated knock-in efficiency has been improved greatly by introducing double-strand breaks at the target sites via engineering endonucleases, it is inefficient in many cells, especially for non-dividing cells [30]. To solve this problem, the non-homologous end joining (NHEJ) pathway instead of HR pathway can be used to integrate foreign gene fragments [32,33,34]. Thus, we can choose alternative knock-in strategy to improve the integration efficiency.

Here, we reported that *GAPDH* can be applied for use as an alternative safe harbor locus in the pig genome. We integrated the GFP gene in this site by CRISPR/Cas9 mediated homologous recombination, while both the exogenous GFP and endogenous *GAPDH* can be expressed independently as normal. Taken together, our study provides a new strategy to knock-in transgene in the pig genome, and we hope our strategy can also be adopted and used in other species.

## Figures and Tables

**Figure 1 genes-10-00660-f001:**
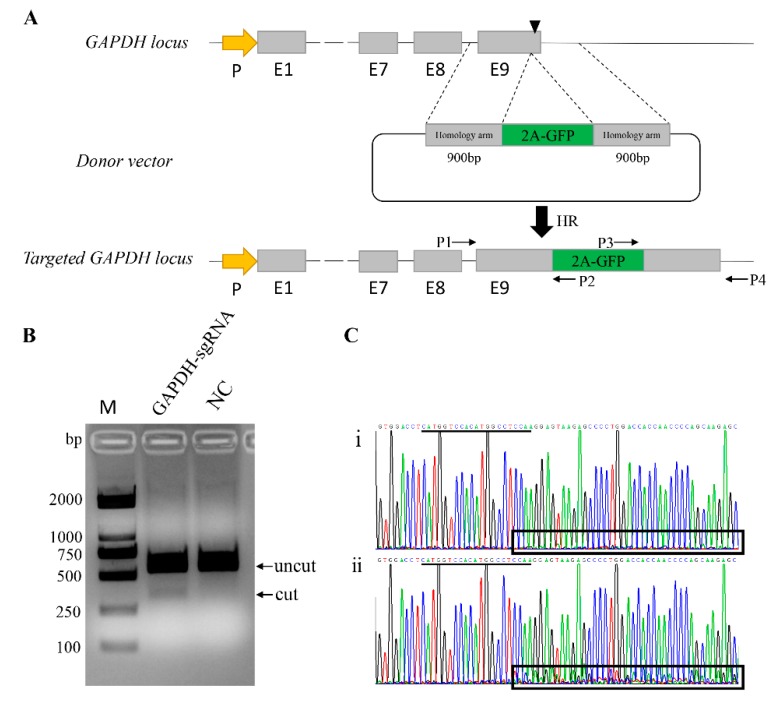
Schematic overview depicting the targeting strategy for the *GAPDH* locus. (**A**) Exons of *GAPDH* are shown as gray boxes, and the black triangle box in Exon 9 represents the sgRNA targeting site. The arrow box in yellow represents *GAPDH* promoter. Targeting vector was created corresponding to the cleavage location of Cas9 and carried each 900 bp regions of homology to the *GAPDH* sequence astride the cleavage site. Two pairs of primers are designed for genotyping. P1/P2 and P3/P4 are designed for genotyping the 5′ and 3′ junctions in the transgene colonies, respectively. (**B**) Identification the activity of GAPDH-sgRNA by T7EN1 cleavage assay. NC, Negative Control; M, Marker, DL2000. (**C**) Sequence analysis showed that the presence of multiple peaks after the targeted site in the sequencing curves clearly distinguishes (i) non-targeted cells from (ii) mutants. sgRNA sequence is underlined in black.

**Figure 2 genes-10-00660-f002:**
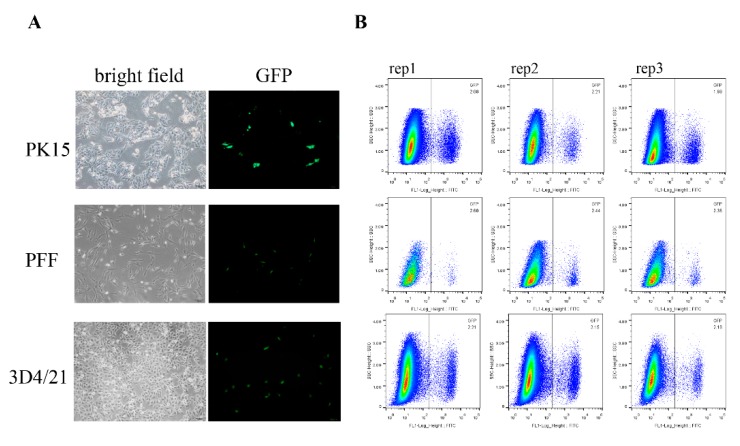
Fluorescence-activated Cell Sorting (FACS) analysis of the GFP knock-in efficiency of different cell types. (**A**) Different cell types were transfected with donor vector and sgRNA expression vectors and then the knock-in efficiencies were visualized by fluorescence microscopy and measured by FACS. Scale bar, 100 μm. (**B**) The GFP knock-in efficiency measured by FACS. The knock-in rates are shown in the upper right corner. Each experiment was repeated three times.

**Figure 3 genes-10-00660-f003:**
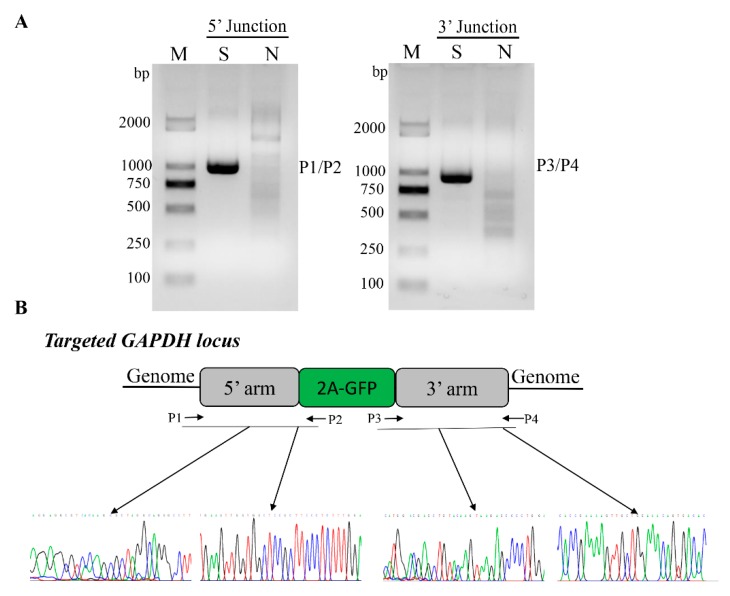
Validation of knock-in events in GFP+ cells. (**A**) PCR genotyping confirmed GFP knock-in of sort cell by FACS. Lane S represents GFP knock in cells sorted by FACS, and lane N represents negative control WT cells. M, Marker, DL2000. (**B**) Nucleotide sequence analysis of junctions between endogenous and exogenous DNA corresponding to HR events. P1/P2 and P3/P4 primers were used to amplify specific region for the left- and right-hand junctions are indicated by dark arrows respectively. Primary nucleotide sequence data corresponding to transition regions between the homology arms of targeting vector and outward host chromosomal DNA, and between the homology arms of targeting vector and inward transgene DNA.

**Figure 4 genes-10-00660-f004:**
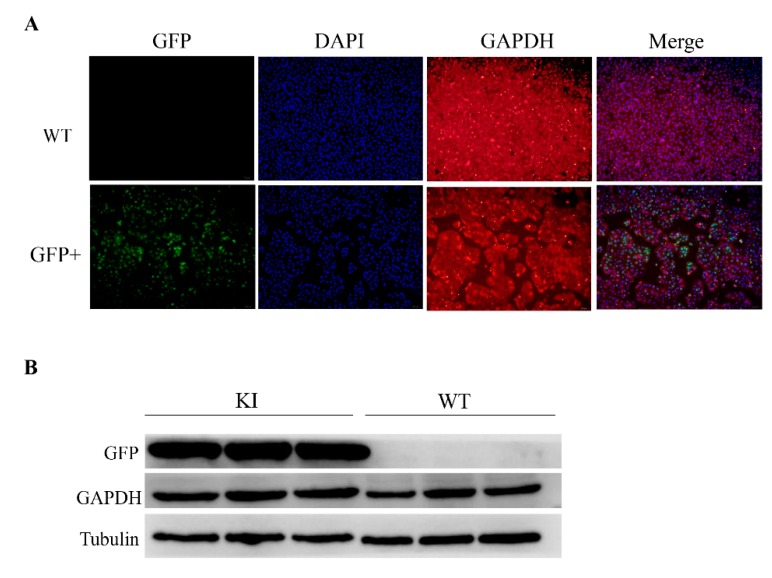
Validation the expression of *GAPDH* in GFP+ cells. (**A**) Immunofluorescence Assay (IFA) analysis of the expression of GFP and *GAPDH* in GFP knock-in cells. Scale bar, 50 μm. WT, wild type control cells; GFP+, GFP knock-in cells. (**B**) Western blot analysis confirmed GFP and *GAPDH* expression in the GFP+ PK15 cells (n = 3). KI, knock-in; WT, wild type.

**Table 1 genes-10-00660-t001:** Analysis of potential off-target sites.

#	Predicted OTS	Sequence	Indel
	GAPDH-sgR	CATGGTCCACATGGCCTCCA AGG	
1	Prediated-OFF-Target1	CATGGTCCCCATGGCCTGCC TGG	NO
2	Prediated-OFF-Target2	CATGATCCGCATGGCCTCCA TGG	NO
3	Prediated-OFF-Target3	CACGGTCCACATGGCCTCCC TGG	NO
4	Prediated-OFF-Target4	CATGGTCTCCATGGCCTCCA GGG	NO
5	Prediated-OFF-Target5	CATGGTGAACATGTCCTCCA TGG	NO
6	Prediated-OFF-Target6	GATGCTCCACCTGGCCTCCA GGG	NO
7	Prediated-OFF-Target7	CAGGGTCCAGATGGTCTCCA GGG	NO

Seven predicted off-target sites were picked up, and PCR-sequencing was performed to identify the results. The PAM sequences are underlined; red highlight letters marked the difference of the sgRNA with the target sequence.

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
