# Peer review of "Identification of Glyceraldehyde-3-Phosphate Dehydrogenase Gene as an Alternative Safe Harbor Locus in Pig Genome"

_genes, 2019, doi:10.3390/genes10090660_

Round 1

Reviewer 1 Report

The purpose of this study is finding another integration locus in pig genome for CRISPR-Cas9 knock-in. The authors hypothesized that 3’-end of house-keeping genes (HKGs) such as GPADH might be a proper safe integration locus because of the following two reasons. First, generally HKGs’ transcription level is pretty high. Second, C-terminal fusion protein of interest through 2A cleavable peptide does not interfere the function of N-terminal HKGs. The data showed that the authors’ hypothesis worked as they thought. The only drawback of this study was RT-PCR is the only data point to prove the GAPDH integration didn’t have any malicious impact on the pig cells. The additional analysis such as growth properties or disease vulnerability will be required to validate the “safe” integration completely. However, the current analysis is enough to at least show that the authors’ hypothesis worked without any urgent severe negative phenotype.

Overall, the manuscript is well written with clear figure presentations. This reviewer noticed the following minor points.

All p11 locus, H11 locus in the manuscript should be pH11 locus. Please keep its consistency.

Line135: HPR might be HRP, horse radish peroxidase.

Reviewer 2 Report

The authors tried to prove that GAPDH is a safe harbor locus in the pig genome. However, they did not give any scientific ground or reason anywhere to say why GAPDH was chosen as a target for investigation. Based on their approach by using 2A-GFP to knock-in at the end of a gene, literally any gene will give the same results. 

Moreover, the authors drew the conclusion of GAPDH is a safe harbor locus based on only the results of GAPDH protein expression is maintained after Knockin is an overstatement. In order to qualify as a safe harbour locus, it requires the integration (knock-in) do not cause alterations of the host genome posing a risk to the host cell or organism. In that case, more analyses (e.g. RNA-seq to show that no effect at the transcriptomic level globally in the Knock-in sample vs the wild-type) has to be done before GAPDH can be claimed as a safe harbor locus.

Minor points:

1) Figure 1B: Missing targeting efficiency below the gel as mentioned in the legend.

2) Methods 2.6: Dilution of antibodies for Western should be included.

3) Figure 4A: Scale bar should be included.

Reviewer 3 Report

This manuscript by Han and co-workers is reasonably well written (although there are some language issues, see below for details). It characterizes a CRISPR/Cas9-based tool which will potentially be useful for pig genetics research and pig breeding. The overall structure of the manuscript makes sense, but the rationale of some experiments as well as several technical details need to be explained more clearly for this paper to reach its full potential. There are also a few language issues.

Here, the authors characterize the pig GAPDH locus as a safe harbor locus for transgenic expression of GFP. This serves as a proof of principle to enable the expression of other constructs at this locus. The overall strategy is straight-forward, but one key feature of their construct is not explained at all, and that is the use of 2A-GFP. The inclusion of the self-cleaving 2A peptide is very important here, and this requires an explanation to ensure that the readers understand that GAPDH and GFP will be transcribed from the same promoter, but will be translated into separate proteins. A less-informed reader might mistake their strategy as an attempt of tagging GAPDH with GFP (this could be done in line 152 after the description of the targeting towards the 3’ end of GAPDH and/or in line 275).

Specific comments:

22-23: sentence “However, this is unappeasable to investigate gene function in pig genome.” does not make any sense.

24-26: better “Here, we provide a new strategy for efficient transgene knock-in in the endogenous GAPDH gene via CRISPR/Cas9 mediated homologous recombination with a minor damage.”, also what do they mean by minor damage? That Cas9 introduces a double-stranded DNA break to promote homologous recombination, or that mutations are introduced in the process?

26-29: better “This strategy has no influence on the expression of the endogenous GAPDH gene. Thus, the GAPDH locus is a new alternative safe harbor locus in the pig genome for foreign gene knock-ins. This strategy is promising for agricultural breeding and biomedical model applications.” The current version talking about “the expression of inner genes” is misleading, because it would mean that the authors looked at the expression of ALL the genes in the pig genomes to test whether their transgenic integration at GAPDH affect this. They did not do this, what they checked is the expression of GAPDH with and without the integrated GFP.

52-63, 130-136, 148-155 (see below), 256-257 need proper language editing.

66: “One pair of sgRNA” does not make sense, do you mean oligonucleotides which will form the sgRNA?

69: “Zhang at the Broad Institute” needs a reference

119: Unclear meaning of “One day before”, one day before what?

Section 2.5 lacks technical specifics of the microscope and camera, what objectives, filters, software steering the microscope/camera, etc.

Section 2.6 needs to describe how HRP was detected on the blots.

Figure 1B: percentage of indel (mutations?) is not given under the gel image (as stated in the legend). Figure legend should identify this clearly as the T7EN1 assay.

Figure 1B & 180: unclear why mutations suddenly are called indels?

Figure 2 shows microscopical and flow cytometry analysis (not FACS), legend is rather badly written and very confusing.

225: “marker” not “maker” needs to be defined.

Corrections throughout the manuscript:

Please ensure that all gene names are italicized, and that other formatting of gene names is also consistent.

Sometimes there are spaces missing between a word and a bracket, e.g. line 37.

ensure that sub- and superscripts are correctly formatted, e.g. CO2 in line 86, AriaTM in line 114.

When referring to companies as suppliers of materials, not only the company name, but also City and Country of company headquarters need to be provided.

“homology arms”  and “left/right arm” for flanking homologous sequences is lab jargon and should be rephrased.

Minor corrections:

21: better “that two loci (Rosa26 and pH11) were identified as safe harbor loci in the pig genome”

23-24: better “Moreover, integration of a transgene may interfere with the endogenous gene expression of the target locus after the foreign fragments are inserted.”

35: better “of foreign genes in a genome is randomly performed”

41: better “and laborious as construction of donor vectors is required”

47-48: better “a Cas9 endonuclease. The Cas9 protein can be introduced to the target locus under the guidance of the sgRNA. Their high efficiency and simple constructs”

51: remove “clinical relevance”

121: RT is not defined, I presume room temperature?

144: better “whether any mutations occured”

148-155: better “This study aims to identify whether the endogenous GAPDH gene can act as an alternative safe harbor locus in the pig genome. Thus, we constructed a reporter system targeting the GAPDH locus in the pig genome. One sgRNA was designed to target the 3’ end of GAPDH just upstream of the stop codon. To achieve GFP knock-in at the GAPDH locus, a homologous recombination vector was constructed as a donor vector (Figures 1A, S1). The donor vector was generated to carry a promoterless 2A-GFP sequence flanked by two regions of homology by modifying the pCDNA3.1(+) vector. When HR-mediated knock-in events occurred, the 2A-GFP fragment was inserted in frame with the endogenous GAPDH coding sequence and resulted in GFP expression”

171: “BbsI” not “Bbs1”

176: “negative control cells” not “group”

185: better “showed that no mutations occurred in the potential off-target loci.”

202: donor plasmid needs to be defined

204-205: rather confusing, just remove “as well as PK15 cells,”

220, 267, 269: better “the 3’ end of the GAPDH gene”, downstream is not a noun and works only in relation to a defined point

258: better “promoter needs to be added”

274: “such as” not “including”, remove “and so on”

283: better “the exogenous GFP and endogenous GAPDH can be expressed independently as normal”

Supplementary materials: Figure S2 is called S1 in the legend erroneously

Reviewer 4 Report

In this manuscript the authors describe the genomic targeting of GAPDH locus with an in-frame 2A-GFP using CRISPR/cas9 technology in Pig cells. The manuscript can be accepted for publication provided the following concerns are addressed.

The authors engineer pig cells with endogenous GAPDH fused to 2A-GFP reporter. While the targeting seems to have gone as per plan, it is unclear whether GAPDH enzymatic activity was perturbed. The authors must perform a GAPDH activity assay to demonstrate that GAPDH-GFP performs equivalently in targeted and untargeted cells. In Fig. 1B legend, the authors say that ratio values are listed under the gel when no such values have been reported in the manuscript. This should be corrected. In Fig. 2A, scale bars are missing from the panels. In 2B, the font size for numbers should be bigger. In Fig. 4A, the resolution of the GAPDH staining panel could be improved.

Round 2

Reviewer 2 Report

The authors have addressed all my suggestions. However, please provide references to support the following statement in the introduction to back up the rationale used in this study.

Line 56-57:

Many house-keeping genes are used as reference genes in many molecular techniques due to their stable and high expression (references required).

Author Response

Thank you very much for point out this question. We have added a reference in the end of this sentence. 

Line 56-57: Many house-keeping genes are used as reference genes in many molecular techniques due to their stable and high expression (18).

Reviewer 4 Report

The authors have addressed all the concerns appropriately. The manuscript is acceptable for publication.

Author Response

Thank you very much, we appreciate for your positive comments on our work.